# The Circadian Clock in the Retinal Pigment Epithelium Controls the Diurnal Rhythm of Phagocytic Activity

**DOI:** 10.3390/ijms23105302

**Published:** 2022-05-10

**Authors:** Christopher DeVera, Jendayi Dixon, Micah A. Chrenek, Kenkichi Baba, Yun Z. Le, P. Michael Iuvone, Gianluca Tosini

**Affiliations:** 1Department of Pharmacology & Toxicology and Neuroscience Institute, Morehouse School of Medicine, Atlanta, GA 30310, USA; chris.devera@nih.gov (C.D.); bkenkichi@msm.edu (K.B.); 2Department of Ophthalmology and Emory Eye Center, Emory University School of Medicine, Atlanta, GA 30322, USA; jendayi.azeezah.dixon@emory.edu (J.D.); micah.chrenek@emory.edu (M.A.C.); miuvone@emory.edu (P.M.I.); 3Departments of Medicine, Cell Biology, and Ophthalmology and Harold Hamm Diabetes Center, University of Oklahoma Health Sciences Center, Oklahoma City, OK 73104, USA; yun-le@ouhsc.edu; 4Department of Pharmacology, Emory University School of Medicine, Atlanta, GA 30322, USA

**Keywords:** retinal pigment epithelium, retina, phagocytosis, aging, circadian clocks, *Bmal1*, transcriptome

## Abstract

The diurnal peak of phagocytosis by the retinal pigment epithelium (RPE) of photoreceptor outer segments (POS) is under circadian control and believed that this process involves interactions from the retina and RPE. Previous studies have demonstrated that a functional circadian clock exists within multiple retinal cell types and RPE. Thereby, the aim of this study was to determine whether the clock in the retina or RPE controls the diurnal phagocytic peak and whether disruption of the circadian clock in the RPE would affect cellular function and the viability during aging. To that, we generated and validated an RPE tissue-specific KO of the essential clock gene, *Bmal1*, and then determined the daily rhythm in phagocytic activity by the RPE in mice lacking a functional circadian clock in the retina or RPE. Then, using electroretinography, spectral domain-optical coherence tomography, and optomotor response of visual function we determined the effect of *Bmal1* removal in young (6 months) and old (18 months) mice. RPE morphology and lipofuscin accumulation was determined in young and old mice. Our data shows that the clock in the RPE, rather than the retina clock, controls the diurnal phagocytic peak. Surprisingly, absence of a functional RPE clock and phagocytic peak does not result in any detectable age-related degenerative phenotype in the retina or RPE. Thus, our results demonstrate that the circadian clock in the RPE controls the daily peak of phagocytic activity. However, the absence of the clock in the RPE does not result in deterioration of photoreceptors or the RPE during aging.

## 1. Introduction

Circadian clocks are present in almost all tissues and cells throughout the body [1], and the molecular mechanism responsible for the generation of circadian rhythms consists of two transcriptional translation feedback loops (TTFL) involving several genes (i.e., clock genes) and their protein products [1]. Removal of clock gene *Bmal1* (i.e., *Arntl*) abolishes circadian rhythmicity at the behavioral and cellular level [2] and induces premature aging, reduced life span, astrogliosis in the brain, and organ shrinkage [3,4].

The mammalian eye also possesses a complete circadian system that controls many physiological functions within this organ [5], and a few studies have also shown that *Bmal1* plays an important role in the maintenance of ocular health. Specifically, germline *Bmal1* knockout (KO) mice show an increased incidence in cataract, corneal inflammation [3], and reduced photoreceptor viability during aging [6]. The circadian regulation of the retinal transcriptome is also dramatically affected by *Bmal1* removal since only a few genes—out of the thousand genes that are rhythmically expressed in wild-type mice—show some rhythmicity [7]. Finally, in *Bmal1* KO mice the circadian regulation of the photopic electroretinogram (ERG) is also absent [7]. These same results are also observed in mice lacking *Bmal1* only in the neural retina [7,8]. Retinal specific *Bmal1* KO mice show alterations in retinal circuitry, and cone viability is significantly reduced during the aging process [8]. Additionally, recent studies have reported that *Bmal1* regulates the spatial expression of cone opsins in mouse retina through the activation of the gene encoding thyroid hormone deiodinase 2 (*Dio2*) [9], and regulates retinal neurogenesis [10].

The retinal pigment epithelium (RPE) is constituted as a monolayer of post-mitotic epithelial cells that play an important role in the maintenance of photoreceptor health and function [11]. Several studies have shown that the phagocytosis of the rod photoreceptor outer segment (POS) discs shows a daily peak occurring one to two hours after onset of light [12,13,14], persists in constant darkness [12,13,15] and does not depend on the brain’s master circadian clock (i.e., the suprachiasmatic nuclei of the hypothalamus) [16,17]. Consistently with these earlier results, more recent studies have demonstrated that the RPE possesses a functional circadian clock [18,19]. A previous study reported that in vivo disruption of the daily peak in phagocytic activity in beta-5 integrin KO mice has negative consequences on the health of the RPE and photoreceptors during aging [20], although more recent work suggests that the lack of the peak may not be detrimental for the health of the RPE and photoreceptors [21,22]. In our previous study, we also reported that although the daily rhythm in phagocytic activity in the RPE of dopamine D_2_ receptor (D_2_R) KO mice is no longer present, clock gene expression in the RPE is not affected by the removal of D_2_R signaling [21]. Therefore, we inferred that the lack of a negative phenotype was probably due to a persistent circadian regulation of RPE function.

The aim of the current study was to first determine whether the circadian clock located in the RPE drives the diurnal peak in the phagocytosis of POS by the RPE and then to determine whether disruption of the circadian clock in the RPE would negatively affect the health of RPE and of the photoreceptors during aging.

## 2. Results

### 2.1. Removal of Bmal1 from RPE Cells

RPE^cre^; *Bmal1^fl/fl^* mice were feed from post-natal day 60 to 74 (p60 to p74) with dox diet (i.e., RPE Bmal1 KO) or control diet (i.e., RPE Bmal1 WT); Figure 1A. Flat mount from an RPE Bmal1 WT mouse crossed with an Ai6(RLC-ZsGreen) cre reporter mouse model demonstrated about 95% cre recombinase expression (Figure 1B) with a majority of expression in RPE cells, but with rare cre recombinase expression in the plexiform layers as demonstrated in a transverse eye section (Figure 1C). Bmal1 immunoreactivity was abolished in the majority of the RPE cells (Figure 1E) and quantification of Bmal1 by Western blot indicated that Bmal1 levels were reduced by more than 70% in RPE Bmal1 KO when compared to RPE Bmal1 WT mice (Figure 1F–G, unpaired *t*-test, *p* < 0.05).

### 2.2. Loss of Bmal1 in the RPE Reduces the Daily Peak of Phagocytic Activity by the RPE

Figure 2A shows a representative immunocytochemical image of rhodopsin-positive phagosomes in the RPE. A clear daily pattern was present in the control (Figure 2B, *p* < 0.05, two-way ANOVA) whereas deletion of *Bmal1* from the RPE blunted the daily diurnal phagocytic peak at ZT1 when compared to ZT23 and ZT3 (Figure 2C, *p* > 0.1, two-way ANOVA, Tukey post hoc). The nocturnal phagocytic peak at ZT14 was not affected by *Bmal1* removal from the RPE when compared to both ZT11 and ZT17 (Figure 2C, # = *p* < 0.05, two-way ANOVA, Tukey post hoc). Wild-type controls for the retina maintained both the morning phagocytic peak (Figure 2D, * = *p* < 0.05, two-way ANOVA, Tukey post hoc) and nocturnal phagocytic peak (Figure 2D, # = *p* < 0.05, two-way ANOVA, Tukey post hoc). Removal of *Bmal1* from the retina did not affect the diurnal peak of RPE phagocytic activity at ZT1 when compared to ZT23 and ZT3 (Figure 2E, * = *p* < 0.05, two-way ANOVA, Tukey post hoc) or the nocturnal peak at ZT14 when compared to ZT11 (Figure 2E, # = *p* < 0.05, two-way ANOVA, Tukey post hoc). Interestingly, there were no differences in the total number of phagosomes engulfed in all mouse models used (Figure 2F; *p* > 0.1, one-way ANOVA). We then investigated the phagocytic activity in the different genotypes at each timepoint sampled. At most times of the day, the greatest differences amongst the three groups were observed for the RPE Bmal1 KO (Figure 2G, * = *p* < 0.05, two-way ANOVA, Tukey post hoc).

We have previously reported that activation of dopamine D2 receptor (D_2_R) signaling is required for the presence of the diurnal phagocytic peak of photoreceptor outer segments [21]. Thus, we decided to investigate whether removal of *Bmal1* from the RPE will also affect D_2_R expression. Consistent with our previous study, D_2_R expression was significantly reduced (about 70%) in RPE Bmal1 KO mice when compared to RPE Bmal1 WT mice when measured at ZT1 (Figure 3A,B; *p* < 0.05, unpaired *t*-test).

### 2.3. Loss of Bmal1 in RPE Alters the Daily Transcriptome

RNA-sequencing (seq) analysis of the RPE Bmal1 KO transcriptome visualized as a heat map (Figure 4B) revealed a different visual signature when compared to RPE Bmal1 WT (Figure 4A). Our RNA-seq analysis detected 20,812 protein coding transcripts (i.e., genes) in the RPE and, consistent with other reports [23,24,25], about 20% of the (4766) genes in RPE Bmal1 WT showed a difference among the different time-points (Figure 4C; *p*. adj. < 0.05; one-way ANOVA, Tukey post hoc). However, in RPE Bmal1 KO samples the number of transcripts showing a difference among the different-time points was reduced to 3526 protein coding transcripts (Figure 4C; *p*. adj. < 0.05; one-way ANOVA, Tukey post hoc). A Venn diagram analysis comparing the rhythmic (r) and non-rhythmic (nr) datasets for each genotype revealed that 830 protein coding transcripts remained rhythmic despite the loss of *Bmal1* in the RPE (Figure 5A^[1]^). Additionally, 3936 protein coding transcripts that were rhythmic in RPE Bmal1 WT samples loss rhythmicity in RPE Bmal1 KO samples (Figure 5A^[2]^). Interestingly, 2696 protein coding transcripts that were non rhythmic in RPE Bmal1 WT samples were now rhythmic in RPE Bmal1 KO samples (Figure 5A^[3]^). In total, 13,350 protein coding transcripts were identified to be non-rhythmic in both genotypes (Figure 5A). To understand the biological pathways that might have been activated in the previously described datasets, a KEGG pathway analysis was performed (Figure 5B–D). The 830 protein coding transcripts that remained rhythmic despite the loss of *Bmal1* in the RPE are associated with cell maintenance, such as morphology, metabolic processes, transepithelial transport processes, and background phagocytic pathways (Figure 5B). Moreover, the 3936 protein coding transcripts that lost rhythmicity in RPE Bmal1 KO samples were previously identified to be involved in mediating the peak in phagocytosis of photoreceptor outer segments and turnover of engulfed outer segments (Figure 5C) [11]. Finally, the 2696 protein coding transcripts that gained rhythmicity in the absence of *Bmal1* did not associate with a particular pattern of activated pathways with only 2 KEGG pathways being statistically significant (Figure 5D).

### 2.4. Removal of Bmal1 from the RPE Does Not Affect Retinal Structure during Aging

Total retina thickness did not differ between the young or old RPE Bmal1 KO compared to RPE Bmal1 WT (Figure 6A,B; *p* > 0.1, two-way ANOVA). No significant differences were observed in the thickness of the photoreceptor layer between the two genotypes and ages (Figure 6A,C; *p* > 0.1, two-way ANOVA). Analysis of the scotopic and photopic ERGs did not indicate any significant differences in the amplitudes of the a- and b-waves between the two genotypes at both ages (Figure 7A–C; *p* > 0.1, two-way ANOVA). No difference among the two genotypes was also observed in the visual acuity and contrast sensitivity at both ages (Figure 8A,B; *p* > 0.1, two-way ANOVA).

### 2.5. Removal of Bmal1 Does Not Produce Morphological Abnormalities in the RPE

Finally, no significant changes in RPE morphological parameters were observed in old RPE Bmal1 KO with respect to age-matched RPE Bmal1 WT mice (Figure 9A–D; *p* > 0.1, unpaired *t*-tests). An autofluorescent particle analysis of the apical membrane of RPE cells (Figure 10A,B) also revealed no significant differences between old RPE Bmal1 KO and RPE Bmal1 WT (Figure 10C, unpaired *t*-test, *p* > 0.1).

## 3. Discussion

The circadian clock system in the eye plays an important role in the modulation of several important ocular functions [5], and the daily and/or circadian regulation of the phagocytic activity of the POS by RPE is probably the most studied of these rhythms [12,13,16,17,20,21,26,27]. However, it is still unclear whether the presence of the phagocytic peak 1–2 h after the onset of light is important for the health of the RPE and photoreceptors and whether this rhythm is controlled by the retinal or RPE circadian clocks or by clocks in both tissues. The findings of the present study indicate that a functional circadian clock in the RPE is necessary and sufficient for the daily peak in phagocytic activity by the RPE (Figure 2), whereas disruption of the retinal circadian clock does not affect this rhythm. Our data also confirmed our previous study [21] by showing that the loss of the diurnal phagocytic peak does not produce a detrimental phenotype in the retina or RPE. Surprisingly, our study also indicated that the lack of a functional circadian clock in the RPE does not have detectible negative consequences on the RPE, even 16 months after the circadian clock was disrupted in these cells.

Interestingly, although cre recombinase expression was very high (96%), the reduction in Bmal1 protein levels was lower (about 70%) and Bmal1 immunoreactivity in the RPE cells of the KO mice showed a mosaic pattern (Figure 1). These data are consistent with a previously published study which reported a similar pattern of expression in this mouse line [28].

A recent study has reported the presence of a second peak in the phagocytic activity soon after the onset of darkness [29] (around ZT14) and, interestingly, the presence of this “nocturnal peak” was still present in RPE and retina Bmal1 KO mice (Figure 2). While the biological significance of this second peak in the nocturnal period is still undetermined, it is noteworthy to mention that our recent work has shown that this peak is not present in D_2_R KOs mice [21]. Loss of this peak in D_2_R KOs mice had no apparent deleterious effects in either the retina or RPE. Finally, our data provide additional support for a key role of dopamine via D_2_R signaling in the regulation of the morning peak in phagocytic activity by RPE cells since the loss of *Bmal1* in RPE cells results in diminished levels of D_2_R protein (Figure 3) and thus providing a potential mechanism to explain the loss of the diurnal phagocytic peak of POS in the RPE Bmal1 KO mice.

Many studies have shown that 10–20% of the mouse transcriptome is under circadian clock control in many tissues/organs [24], including the RPE [23,25]. Not surprisingly, the transcriptional processes under circadian regulation in the RPE include many of the pathways known to be involved in the regulation of phagocytic activity and metabolism, such as actin cytoskeleton remodeling [25,30], integrin signaling [20,21], cAMP signaling [31], protein phosphorylation [11] and mitochondrial electron transport chain [23,32]. An additional study has also reported that mice lacking the clock genes *Period1 and Period2* do not show a circadian rhythm in POS phagocytosis and has suggested that this rhythm is likely to be controlled by the RPE via the regulation of genes coding for neurotransmitters [33]. Our analysis of the daily transcriptomic in the RPE of Bmal1 KO mice and WT mice confirm previous studies in showing that about 20% of the RPE transcriptome is rhythmically regulated (Figure 4). Although this result may appear surprising, previous studies have reported that under LD conditions locomotor activity and gene expression may still be rhythmic even in the absence of a functional circadian clock [2,34]. The effect of *Bmal1* removal on the RPE transcriptome are puzzling, but the KEGG pathway analyses shed some light on our data; many (830) of the protein coding transcripts that remain rhythmic despite loss of *Bmal1* play roles in RPE cell maintenance such as cell morphology, metabolic processes, transepithelial transport processes, and background phagocytic pathways. Hence, it appears that while the removal of Bmal1 from the RPE does indeed affect the rhythmicity of phagocytic processes, the transcription of many cellular functions was still rhythmic, and this can partially explain why the removal of a functional circadian clock from the RPE did not induce any negative phenotype in a post-mitotic and a highly metabolic cell type.

Removal of *Bmal1* negatively affects the function and viability of many cell types in the brain and the eye [3,4,7,8,9,10] and, therefore, our new results are somewhat puzzling. Our circadian transcriptomic analysis demonstrated that despite the reduced expression of Bmal1, RPE cells managed to maintain cellular processes that are important for daily function [11] and can partially explain why we have not observed any detrimental phenotypes in the retina and RPE cells. Additionally, several previous studies have also reported that (i) post-embryonic disruption of *Bmal1* may produce fewer detrimental effects compared to germline *Bmal1* removal [35]; (ii) rhythmic metabolic processes may persist in a tissue even in the absence of *Bmal1* [36]; (iii) the presence of *Bmal1* in 30% of RPE cells may still drive rhythmic gene expression in the entire tissue (Figure 1), (iv) the unchallenging, unnatural housing conditions (e.g., light, food, etc.,) in which the laboratory mice are maintained may mitigate the consequence of a dysfunctional circadian clock [37] and finally (v) the presence of functional circadian clocks in the neighboring cells/tissue may mitigate the negative effect due to the loss of a functional circadian clock in a specific cell type. Further studies will be needed to address this important point and thus to fully understand the role played by the circadian clock in the regulation of RPE functions.

In conclusion, our study using a tissue-specific *Bmal1* KO mouse model demonstrated that the circadian clock in the RPE controls the daily diurnal peak in phagocytosis of POS. In addition, our data also indicate that loss of the diurnal phagocytic peak of POS or a functional RPE circadian clock does not result in apparent deleterious effects in the retina and RPE during aging. Finally, our study also suggests that the disruption of the circadian clock in a tissue-specific manner does not always produce expected negative consequences observed with germline clock gene disruption.

## 4. Materials and Methods

### 4.1. Animals

Rpe-CRE (Vmd2-rtTA/TetO-cre) [23] and *Bmal1^fl/fl^* (Jackson laboratory Cat# 007668) were bred to generate the conditional and inducible Cre recombinase expression in retinal pigment epithelium. Mice hemizygous for the Cre locus and homozygous for *Bmal1^fl/fl^* locus were crossed to *Bmal1^fl/fl^* mice to generate mice that were homozygous for the Bmal1 locus and either hemizygous for the CRE locus (Rpe-CRE; *Bmal1^fl/fl^*) or non-carrier *Bmal^fl/fl^*. The offspring from these mice were used for experimental purposes once genotyped via qPCR for CRE and PCR for *Bmal1*. The following qPCR primer pairs and probes were used to genotype the Cre locus (CRE: Forward: 5′-CATCGCTCGACCAGTTTAGTT-3′, Reverse: 5′-CTGACGGTGGGAGAATGTTAAT-3′, Probe: 5′-CGCAGGTGTAGAGAAGGCACTTAGC(FAM)-3′, off target control: Forward: 5′-AGGCTTGTCACCCTTCTTTC-3′, Reverse: 5′-TAATACTGGTGCTCACAAGGC-3′, Probe: 5′-ATGCTGACATCTTGGGCACAGACA(HEX)-3′) and the following PCR primers for the *Bmal1^fl/fl^* locus (Forward: 5′-TCCTGGTTGGTCCAAGAATATG-3′, Reverse: 5′-CTGACCAACTTGCTAACAATTA-3′; Knockout forward: 5′ACTGGAAGTAACTTTATCAAACTG-3′). Mice that showed systemic knockout of the Bmal1 locus due to leaky expression of the Rpe-CRE construct were not used for experiments or further breeding. Rpe-CRE; Bmal1fl/fl mice were fed either regular mouse lab diet (RPE Bmal1 WT; LabDiet 5001 Cat# 0001319) or doxycycline (dox; RPE Bmal1 KO; dox; Bioserv Cat# S3888) mouse lab diet at p60–p74. Details about the production, maintenance, and genotyping of Chx10cre; Bmal1fl/fl (Retina Bmal1 KO) and floxed control (Retina Bmal1 WT; Bmal1fl/fl; Jackson Laboratory Cat# 007668) are reported in our previous study8. All mice were housed on a 12:12 light:dark cycle with lights on at 7AM [zeitgeber time [ZT]0) and lights off at 7PM (ZT12). All mice used in this study had access to food and water ad libitum. Additionally, RPE Bmal1 WT and RPE Bmal1 KO mice were aged to 6 months (young) and 18 months (old) of age. All animals were housed in the animal facility at Emory University School of Medicine. All experimental procedures were performed in accordance with the NIH Guide on Care and Use of Laboratory Animals and were approved by the Emory University and the Morehouse School of Medicine Animal Care and Use Committees.

### 4.2. Phagosome Counting Assay

Whole eyes were processed for phagosome counting as previously described [21] and labeled with the primary rhodopsin 4D2 antibody (Abcam Cat# ab98887, RRID:AB_10696805). Briefly, whole eyes from RPE Bmal1 WT and RPE Bmal1 KO (male and female, 12–16 weeks of age, *n* = 8–10 eyes/genotype/timepoint) were collected at ZT1, ZT3, ZT5, ZT8, ZT11, ZT14, ZT17, ZT20, ZT23 (ZT0 = 7 AM, ZT12 = 7PM) under room light or low red light (<1 lux) conditions. The enucleated eyes were fixed in 4% paraformaldehyde for at least 3 h and transferred to 30% sucrose (Fisher Cat# S5-500) for cryoprotection. Once sufficiently cryoprotected, eyes were embedded (Fisher Cat# 4585), sectioned at 12 µm, and stored at −20 °C until processed. On the day of processing, goat serum (Vector Cat# S-1000, RRID:AB_2336615) and fragmented goat anti-mouse antibodies (Jackson Immuno Research Cat# 115-007-003, RRID:AB_2338476) were used for antigen blocking before overnight incubation at 4 °C with rhodopsin 4D2 (1:500, Abcam Cat# ab98887, RRID:AB_10696805). Slides were washed and incubated with goat anti-mouse Alexa Fluor 488 (1:1000, Cell Signaling Technology Cat# 4408, RRID:AB_10694704, Danvers, MA, USA) for up to 2 h at room temperature (RT). Slides were washed and counterstained with DAPI (1:500, ThermoFisher Cat# D1306, RRID:AB_2629482, Hampton, NH, USA). Rhodopsin immunopositive phagosomes localized to the RPE layer (confirmed with DAPI) were counted in four independent 150 micron sections of retina with two sections counted on the nasal side of the eye and the other two sections counted on the temporal side of the eye. This counting process was repeated for at least 8 whole eye transverse sections per mouse per genotype per timepoint. Rhodopsin immunostained sections were visualized on a confocal microscope (Zeiss LSM 700 microscope, RRID:SCR_017377).

### 4.3. Western Blot

RPE tissue from RPE Bmal1 WT and RPE Bmal1 KO male and female mice (*n* = 3 mice/genotype) was isolated as previously described [23] then lysed in ice-cold RIPA buffer (Boston Bioproducts Cat# BP-116X) with protease inhibitors. RPE protein homogenates (at least 10 μg/well) were separated on a tris/glycine/SDS 7.5% gradient mini-protean gel (Bio-Rad Cat# 456-1026) and electrophoretically transferred to a nitrocellulose membrane (Bio-Rad Cat# 1704156) using the Transblot turbo system (Bio-Rad Cat# 1704150) for immunoblotting. Membranes were blocked for 1 h at RT on a shaker in 5% BSA (Tocris Cat# 5217) with 0.1% Tween-20 (Bio-Rad Cat# 170-6531). Each membrane was incubated with primary antibodies (D_2_R, Millipore Sigma Cat# AB5084P and Bmal1, Cell Signaling Technology Cat# 14020, RRID:AB_2728705, Burlington, MA, USA) in 5% BSA with 0.1% Tween-20 overnight at 4 °C on a shaker. The membranes were washed three times for five min in 0.1 M TBS and 0.1% Tween-20 on a shaker at RT. A secondary antibody conjugated with HRP (1:10,000; anti-rabbit; Cell Signaling Technology Cat# 7074, RRID:AB_2099233) were incubated at RT on a shaker for 1 h. The membranes were washed four times for five min in 0.1 M TBS and 0.1% Tween-20 on a shaker at RT. The development of the membranes was accomplished with ECL Western blot substrate (ThermoFisher Scientific Cat# 32106) for five min at RT. Protein bands were visualized using Image J (v1.51w).

### 4.4. RPE RNA Library Preparation and Sequencing

RPE Bmal1 WT and RPE Bmal1 KO male and female mice (12–16 weeks of age; *n* = 3 mice/group/timepoint) were euthanized at ZT1, ZT7, ZT13, and ZT19 with lights on at 7AM (ZT0) and lights off at 7PM (ZT12). RPE tissue collection for RNA isolation and RNA pooling was carried out as previously described [23]. Total RNA samples were sent to GENEWIZ from Azenta Life Sciences (South Plainfield, NJ, USA) for both library preparation and high-throughput sequencing. The Illumina Stranded Total RNA Prep, Ligation with Ribo-Zero Plus (Illumina Cat# 200050529, San Diego, CA, USA) was used for library preparation according to the manufacturer’s instructions. The HiSeq4000 Illumina platform was used for high-throughput sequencing as paired-end 150 base-pair (bp) reads at a sequencing depth of 15–20 million reads per sample.

### 4.5. Bioinformatics and Rhythmic Analysis

Each FASTA file from each biological sample was processed for counts per million (CPM) as previously described [24]. Briefly, all FASTA files were mapped to the mouse genome assembly and transcript annotation, mm10, using Bowtie2 (v2.1.0). Count data were generated with HTSeq-count (Pycharm Community Edition, 2016.3.2) to quantify uniquely mapped reads with settings as previously described to generate count data which were then converted to CPM for pathway analysis with CPM values manually calculated to FPKM (Fragments Per Kilobase of exon model per Million reads mapped) for individual gene analysis. Additionally, all datasets following mapping were filtered for highly expressed genes in photoreceptors and choroid from a dataset published and publicly available [38]. Datasets were further filtered for transcripts common to all datasets. To determine whether genes were rhythmic in either RPE Bmal1 WT or RPE Bmal1 KO samples, a one-way ANOVA was performed within each genotype and any transcript that had an adjusted *p*-value < 0.05 was considered rhythmic (adjustment carried out via the Benjamini–Hochberg multiple comparison correction). The transcripts were then subjected to identification for protein coding and these protein coding transcripts were subjected to KEGG (Kyoto Encyclopedia of Genes and Genomes) pathway analysis. The RNA-seq raw and processed data discussed in this publication have been deposited in NCBI’s Gene Expression Omnibus [39] and are accessible through GEO Series accession number GSE191104 (Available online: https://www.ncbi.nlm.nih.gov/geo/query/acc.cgi?acc=GSE191104 (accessed on 3 April 2022)).

### 4.6. Spectral Domain-Optical Coherence Tomography (SD-OCT) and Fundus Imaging

Retinal thickness measurements were performed as previously described [8]. Briefly, mice were anesthetized and a circular scan 0.57 mm from the optic nerve head was made (Phoenix Micron IV). Total retinal and photoreceptor receptor layer thicknesses were measured and quantified using Adobe Photoshop.

### 4.7. Electroretinography (ERG)

Photic and scotopic ERG measurements were performed, as previously described [8]. Briefly, ERG recordings were obtained midday (ZT4-8) from mice dark-adapted overnight and anesthetized with ketamine/xylazine. Each recording epoch was 250 ms and each stimulus flash presented in an UTAS BigShot ganzfield (LKC Technologies, Gaithersburg, MD, USA). For dark-adapted conditions, a series of 5 stimulating flashes that ranged from 0.00039 to 25.3 cd*s/m^2^ with each flash lasting 20 µs. For light-adapted ERG recordings, a steady background-adapting field of 29.76 cd*s/m^2^ was presented to saturate rod photoreceptors. After 10 min of light-adaptation, each step consisted of 60 flashes of 3.14 cd*s/m^2^ that lasts 2 min for a total measurement time of 35 min. The amplitude of each a-wave was measured from baseline to trough and the b-wave from the trough to the peak of the response.

### 4.8. Opto-Motor Response (OMR)

Mice underwent visual psychophysical testing using the OptoMotry device (CerebralMechanics, Inc. Cat# D430) to assess spatial frequency threshold and contrast sensitivity as previously described [8].

### 4.9. RPE Flat Mount Morphology and Autofluorescence Analysis

RPE flat mount analysis was performed as previously [21,40,41]. Briefly, each eye following enucleation was placed in Z-fix (Anatech Cat# 170) for 10 min at room temperature and then rinsed up to five times with 0.01M phosphate-buffered saline. On a microscope slide (VWR Cat# 16004-406), radial cuts around the limbus of the eye were made using spring scissors (WPI Cat# 501235) to remove the anterior segment of the eye (cornea and lens). The remaining eyecup was divided into 4 petals and allowed to be flattened against the microscope slide. The now exposed retina was peeled by using Dumont #5/45 forceps (Fine Science Tools Cat# 11251-35) and RPE flat mounts were placed in a 24 well plate (Fisher Cat# 12565501) with 0.01M phosphate-buffered saline and processed immediately for zonula occludins-1 (ZO-1; 1:500, Millipore Sigma Cat# MABT11) and Bmal1 (Cell Signaling Technology Cat# 14020, RRID:AB_2728705) immunostaining. RPE flat mounts were blocked and then incubated with ZO-1 antibody (1:500, Millipore Sigma Cat# MABT11) overnight at 4 °C. Flat mounts were washed and incubated in Alexa Fluor 488 (1:500, goat anti-rat, Invitrogen Cat# A110006) for up to 2 h on a shaker at room temperature. RPE cell junctions were visualized on a confocal microscope (Zeiss LSM 700 microscope, RRID:SCR_017377) in the 488 nm channel, and autofluorescent particles were visualized in the 568 nm channel. In order to determine the number of autofluorescent particles per RPE cell, a pipeline in Cell Profiler (v 2.2.0; CellProfiler Image Analysis Software, RRID:SCR_007358) was created to: (1) identify RPE cells on the isolated RPE flat mount via ZO-1 staining, (2) identify all autofluorescent particles present on the RPE flat mount, (3) create a mask from the ZO-1 labeling to identify RPE cells, and (4) count the number of autofluorescent particles in each RPE cell. Additionally, RPE cell morphology parameters such as area, eccentricity, solidity, and compactness were measured from step 1 on isolated RPE cells, as previously described [28]. For all RPE morphology analyses, only central measurements were made on each RPE petal (up to 1.0 mm from the optic nerve head).

### 4.10. Statistical Analyses

For all datasets analyzed, tests for assumptions of normality by Shapiro–Wilk and assumptions of equal variance by the F-test was carried out prior to any statistical test is applied to assess for differences. Statistical analyses were carried out with either unpaired *t*-tests, one-way ANOVA, or two-way ANOVA. Tukey post hoc multiple comparisons test was carried out with significance set at *p* < 0.05 for all statistically significant ANOVA results with power >0.8. Unpaired *t*-tests for comparison between two groups was set at *p* < 0.05 with power >0.8. All data were expressed as mean ± SEM (standard error of the mean). GraphPad prism (v9.3.1) was the statistical analysis program used for all analyses unless otherwise stated.

## Figures and Tables

**Figure 1 ijms-23-05302-f001:**
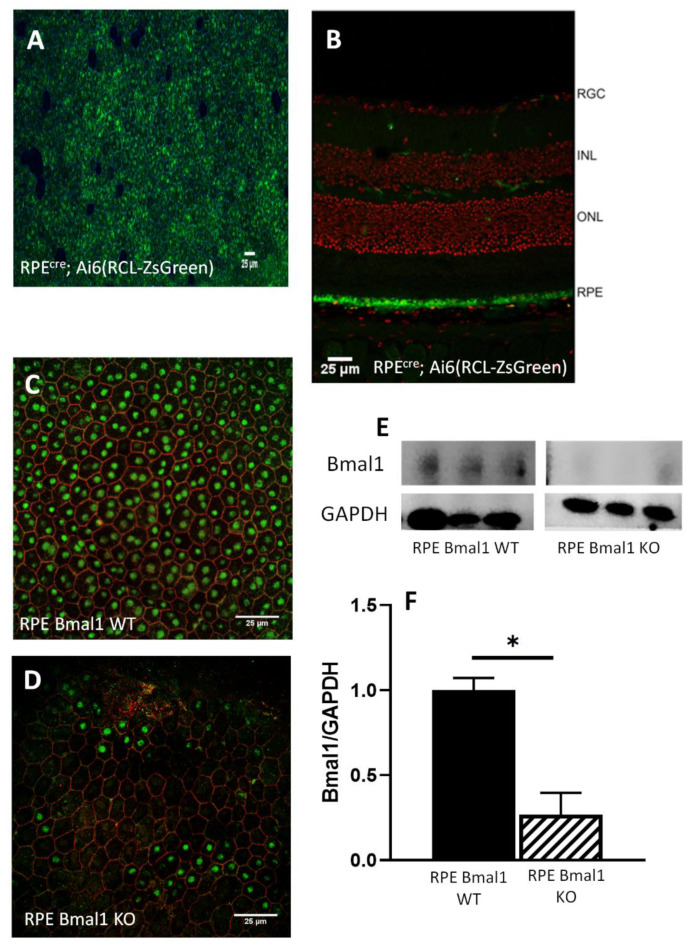
Validation of RPE Bmal1 KO mouse model. RPE cells of RPE^cre^; Ai6(RCL-ZsGreen) mice were visualized *en face* as a flat mount; there was about a 95.8% expression of cre recombinase activity (green) in RPE cells that were demarcated with zonula occludens (**A**; blue). Transverse retinal sections from RPE^cre^; Ai6(RCL-ZsGreen) showed expression of cre recombinase (green) primarily in the RPE cell layer with some unexpected cre recombinase expression in the outer and inner plexiform layer (**B**); the nuclear layers of the retina were stained with propidium iodide (red). Representative microphotographs displaying Bmal1 immunoreactivity (green) in RPE Bmal1 KO and RPE Bmal1 WT mice as prepared in an RPE flat mount with RPE cells demarcated with zonula occludens (red; **C**,**D**). Bmal1 immunoreactivity in RPE cells of RPE Bmal1 KO mice (**D**) was decreased (~70%) compared to expression in RPE Bmal1 WT mice (**C**). To determine the total level Bmal1 protein in the RPE, we measured Bmal1 and GAPDH in total RPE protein extracts by Western blot (**E**,**F**). Bmal1 protein was significantly reduced (about 70%) in RPE Bmal1 KO when compared to RPE Bmal1 WT mice (**F**; * = *p* < 0.05, unpaired *t*-test, *n* = 3 mice/group). Data are expressed as mean ± SEM.

**Figure 2 ijms-23-05302-f002:**
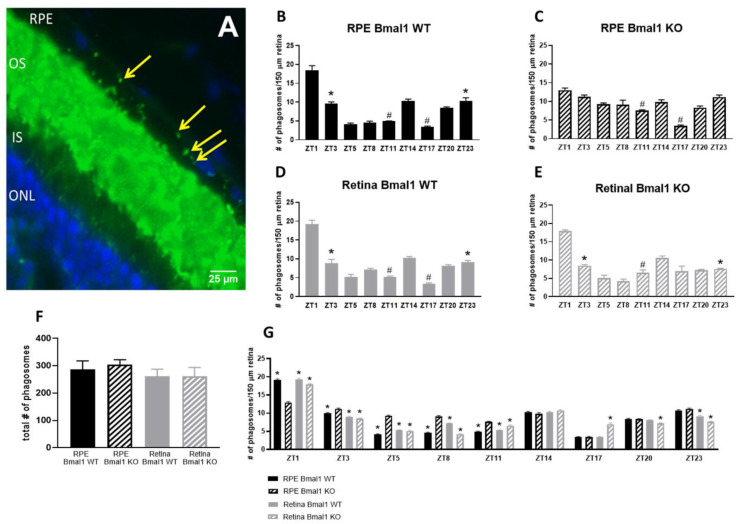
The RPE circadian clock controls the diurnal peak in phagocytosis of POS. One well known function of the RPE is its ability to phagocytose photoreceptor outer segments where a peak in phagocytic activity occurs 1–2 h after onset of light. We wanted to determine whether this function was under circadian regulation, so we used a conditional and inducible Bmal1 knockout in RPE cells. RPE Bmal1 WT, RPE Bmal1 KO, Retina Bmal1 WT, and Retinal Bmal1 KO mice were collected at the following time points: ZT1, ZT3, ZT5, ZT8, ZT11, ZT14, ZT17, ZT20, ZT23 with ZT0 at 7AM (*n* = 4–6 eyes/genotype/time point). A representative microphotograph of a transverse eye section immunostained with anti-rhodopsin antibody (Rho4D2, green) and DAPI (blue) (**A**). Phagosomes (marked with yellow arrows) can be seen as small green particles present in the RPE cell layer (arrows). Bar graphs indicated the number of phagosomes per 150 μm of RPE at different time points in four different genotypes (**B**–**E**,**G**). We then performed various phagocytic peak analyses to determine whether ablation of Bmal1 in the retina or RPE has any effect on the phagocytic capacity of RPE cells. The diurnal phagocytic peak at ZT1 was dampened in RPE Bmal1 KO mice (**C**) while maintained in RPE Bmal1 WT, Retina Bmal1 WT and Retinal Bmal1 KO mice when compared to ZT23 and ZT3 (**B**,**D**,**E**; * = *p* < 0.05, two-way ANOVA, Tukey post hoc). Interestingly, the nocturnal peak observed at ZT14 was maintained in RPE Bmal1 WT, RPE Bmal1 KO, and Retinal Bmal1 WT when compared to ZT11 and ZT17 (**B**,**C**; # = *p* < 0.05, two-way ANOVA, Tukey post hoc). However, this nocturnal peak was only partially observed in Retinal Bmal1 KO mice when compared to ZT11 (**E**; # = *p* < 0.05, two-way ANOVA, Tukey post hoc) and not ZT17 (**E**; *p* > 0.1, two-way ANOVA, Tukey post hoc). No difference was observed in the total number of phagosomes engulfed through the 24 h among the different genotypes (**F**; *p* > 0.1, one-way ANOVA). We then did a group analysis that assessed the phagocytic capacity of Retinal Bmal1 WT, Retinal Bmal1 KO, and RPE Bmal1 WT when compared to RPE Bmal1 KO. Interestingly, all three genotypes demonstrated different phagocytic capacity when compared to RPE Bmal1 KO mice in particular during the daytime period (**G**; * = *p* < 0.05, two-way ANOVA, Tukey post hoc). Data are expressed as mean ± SEM. OS = outer segments; IS = inner segments; ONL = outer nuclear layer.

**Figure 3 ijms-23-05302-f003:**
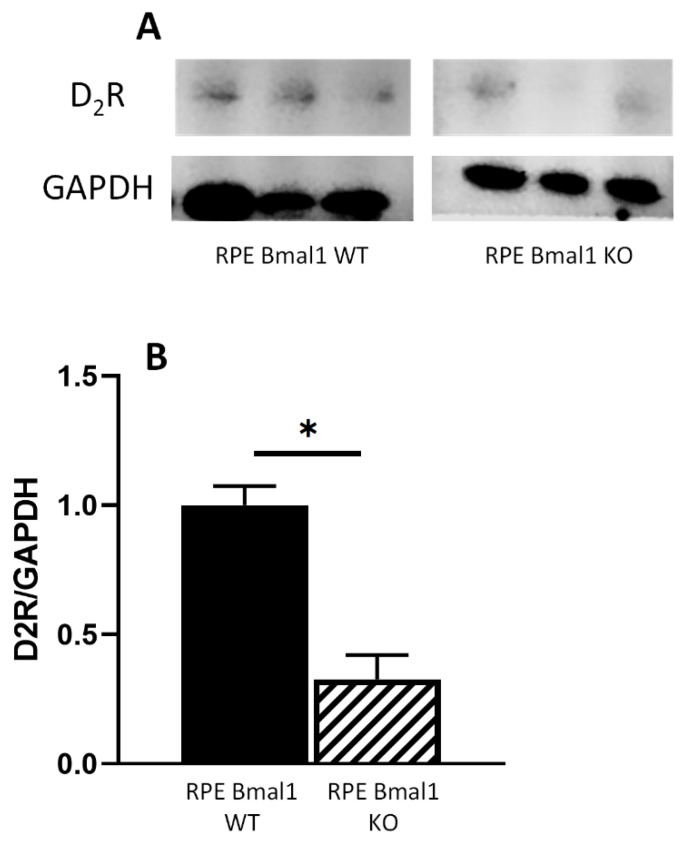
Dopamine 2 receptor protein is decreased in RPE Bmal1 KO mice. We previously identified that dopamine 2 receptor (D_2_R) played a role in mediating the peak in RPE phagocytic activity, leading us to examine the levels of D_2_R protein levels in RPE Bmal1 WT and RPE Bmal1 KO mice. RPE samples were collected at ZT1 (ZT0 at 7AM) and assessed via Western blot analysis. Representative Western blot bands for D_2_R and GAPDH antibodies (**A**). Densitometry analysis of the band intensities indicated a significant reduction in D_2_R protein in RPE Bmal1 KO mice when compared to RPE Bmal1 WT mice (**B**; * = *p* < 0.05, unpaired *t*-test, *n* = 3 mice/group). Data are expressed as mean ± SEM.

**Figure 4 ijms-23-05302-f004:**
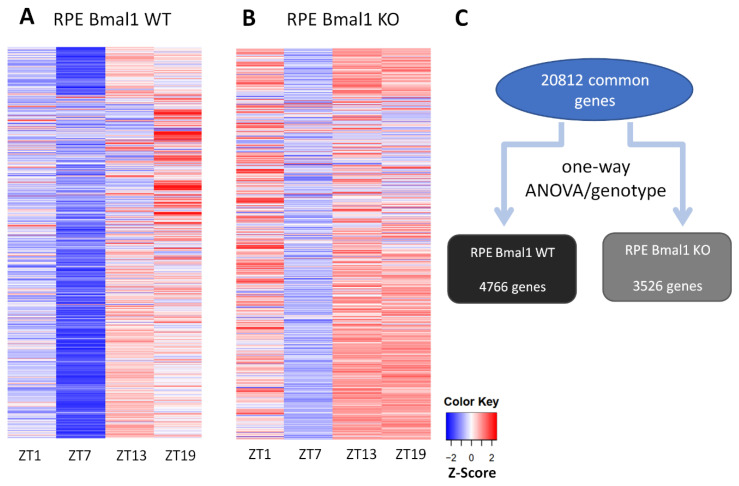
Effect of *Bmal1* removal of the RPE daily transcriptome. RNA-sequencing (-seq) analysis was carried out on RPE samples of RPE Bmal1 WT and RPE Bmal1 KO mice (12–16 weeks of age; *n* = 3 mice/group/timepoint) to assess the role of the RPE clock in rhythmic gene expression. RPE samples were collected at ZT1, ZT7, ZT13, ZT19 with ZT0 at 7AM and high throughput RNA-seq analysis was carried out. Individual transcripts were assessed following mapping to the mouse genome (mm10); the mapped transcripts were quantified as fragments per kilobase of exon per million mapped fragments (FPKM). Using the newly calculated FPKM values, all transcripts common to all datasets were mapped as a heatmap for each genotype across the time points sampled (**A**,**B**). Visually, there was a difference in the z-score distribution of the FPKM values when comparing RPE Bmal1 WT (**A**) and RPE Bmal1 KO (**B**). A transcriptional variance analysis was performed within each genotype across the time points and demonstrated that 4766 genes (about 20%) in RPE Bmal1 WT showed daily variation across the four time points, while the number of genes that varied across the different time points was decreased to 3526 genes in RPE Bmal1 KO samples (**C**; adj. *p*-value < 0.05, one-way ANOVA, Tukey post hoc). The RNA-seq raw and processed data discussed in this publication have been deposited in NCBI’s Gene Expression Omnibus and are accessible through GEO Series accession number GSE191104 (Available online: https://www.ncbi.nlm.nih.gov/geo/query/acc.cgi?acc=GSE191104 (accessed on 3 April 2022)).

**Figure 5 ijms-23-05302-f005:**
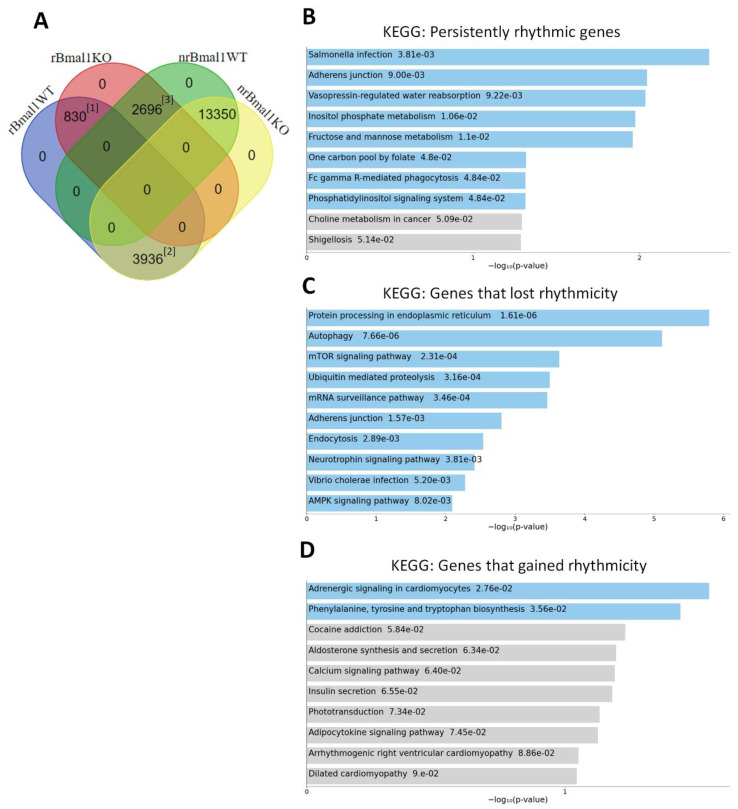
Effect of *Bmal1* removal on the transcription of daily biological pathways. Isolated RPE samples were collected at different time points (ZT1, ZT7, ZT13, ZT19; ZT0 = light onset at 7AM) were subjected to high throughput RNA-sequencing analysis and a rhythmic analysis performed within each genotype. Taking the rhythmic (r) and non-rhythmic genes (nr) datasets, a Venn diagram analysis was carried out (**A**). This comparison analysis revealed that 830 protein coding transcripts remained rhythmic despite the loss of *Bmal1* in the RPE (**A**^[1]^). Additionally, 3936 protein coding transcripts that were rhythmic in RPE Bmal1 WT samples loss rhythmicity in RPE Bmal1 KO samples (**A**^[2]^). Interestingly, 2696 protein coding transcripts that were not rhythmic in RPE Bmal1 WT samples were now rhythmic in RPE Bmal1 KO samples (**A**^[3]^). A total of 13,350 protein coding transcripts were identified to be non-rhythmic in both genotypes (**A**). KEGG pathway analysis indicated that the 830 protein coding transcripts are involved in RPE cell maintenance such as cell morphology, transepithelial transportation, metabolic processes, and background phagocytic processes (**B**); 3936 protein coding transcripts that lost rhythmicity in RPE Bmal1 KO are possibly involved in mediating the peak in phagocytosis of photoreceptor outer segments and turnover of engulfed outer segments (**C**); 2696 protein coding transcripts that gained rhythmicity in the absence of *Bmal1* did not indicate any particular pattern of activated pathways with only 2 KEGG pathways being statistically significant (**D**).

**Figure 6 ijms-23-05302-f006:**
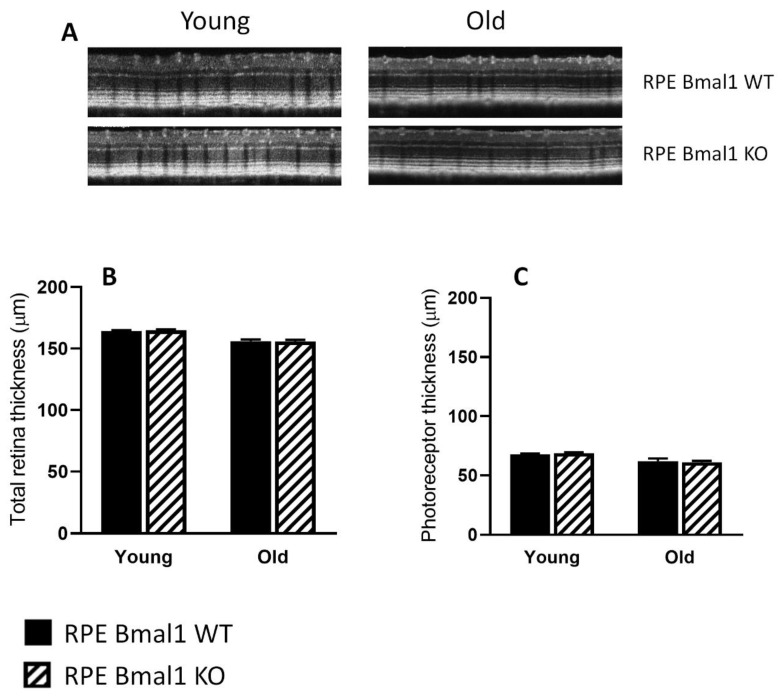
Retinal thickness is not affected by disruption of the RPE circadian clock. Retina thickness was assessed to determine if there were any age-related effects that could be observed with a circadian disruption in the RPE. Representative retinal images taken with SD-OCT from young and old RPE Bmal1 WT and RPE Bmal1 KO mice (**A**). Total retinal thickness was not affected in the RPE of mice lacking *Bmal1* in the RPE at any age measured (**B**; *p* > 0.1, two-way ANOVA, Tukey post hoc, *n* = 4–6/group). No differences were observed in the thickness of the photoreceptor layer between the two genotypes at both ages (**C**; *p* > 0.1, two-way ANOVA, *n* = 4–6 mice/group). Data are expressed as mean ± SEM.

**Figure 7 ijms-23-05302-f007:**
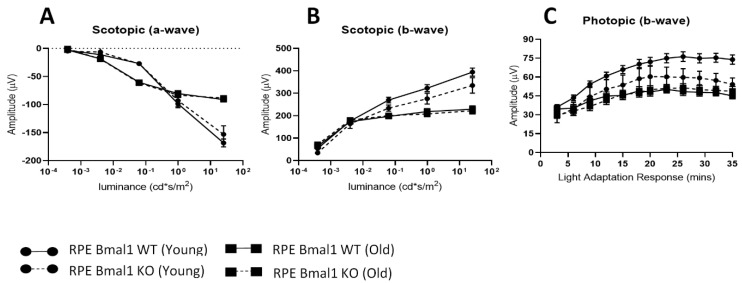
Functioning of rods and cones are not affected by the loss of *Bmal1* in the RPE. Retina function was assessed with electroretinography at the level of both rod (i.e., scotopic) and cone (i.e., photopic) photoreceptors. At both ages examined, there was no difference in rod photoreceptor function in RPE Bmal1 KO when compared to RPE Bmal1 WT (**A**,**B**; *p* > 0.1, two-way ANOVA, *n* = 4–6/group). Additionally, cone photoreceptor function was also not affected in RPE Bmal1 KO when compared to RPE Bmal1 WT mice at all ages (**C**; *p* > 0.1, two-way ANOVA). Data are expressed as mean ± SEM.

**Figure 8 ijms-23-05302-f008:**
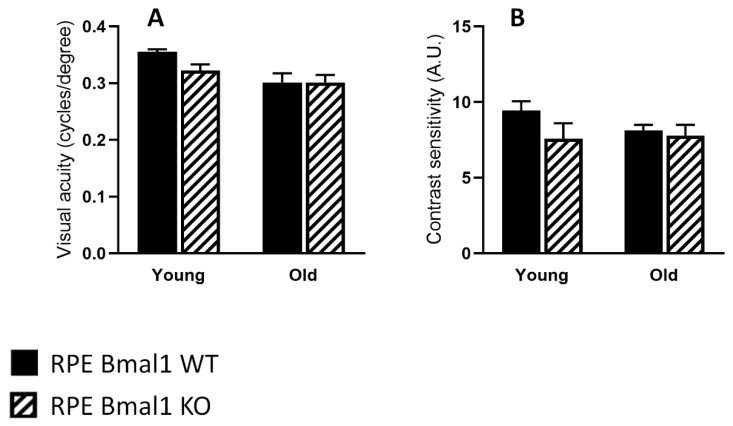
Psychophysical measurements of visual function are not affected by the absence of *Bmal1* in RPE cells. Visual function was assessed with an optomotor response protocol. No differences were observed in visual acuity (**A**, *p* > 0.1, two-way ANOVA) in RPE Bmal1 KO mice when compared to similarly aged RPE Bmal1 WT mice. Contrast sensitivity was also not affected in RPE Bmal1 KO when compared to RPE Bmal1 WT at all the ages tested (**B**; *p* > 0.1, two-way ANOVA, *n* = 4–6/group). Data are expressed as mean ± SEM.

**Figure 9 ijms-23-05302-f009:**
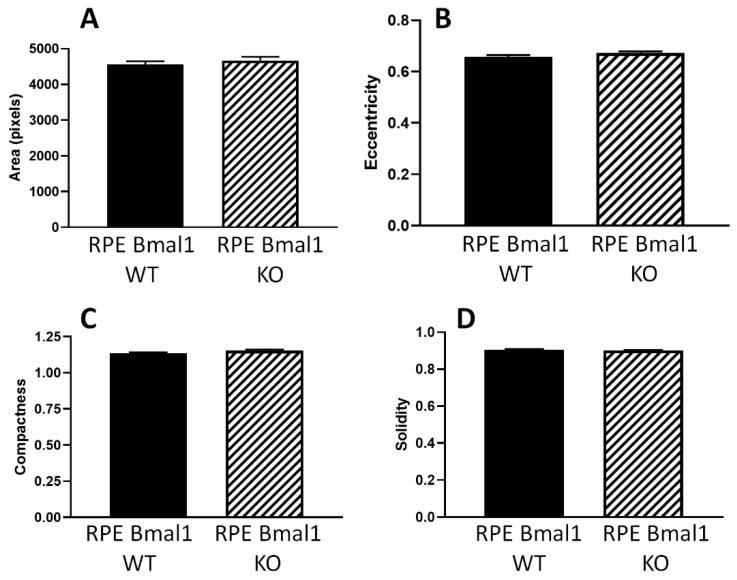
RPE morphology is not affected with loss of *Bmal1* in the RPE. Because RPE cells have a unique cell morphology, we were interested to determine if cell shape parameters would change as a result of prolonged circadian disruption. Using cell profiler (v2.2.0), we were able to measure the area, eccentricity, compactness, and solidity of RPE cells. No differences were observed in any of the four shape parameters measured in old RPE Bmal1 KO mice compared to RPE Bmal1 WT mice (**A**–**D**; *p* > 0.05, unpaired *t*-test, *n* = 5–6/group). Data are expressed as mean ± SEM.

**Figure 10 ijms-23-05302-f010:**
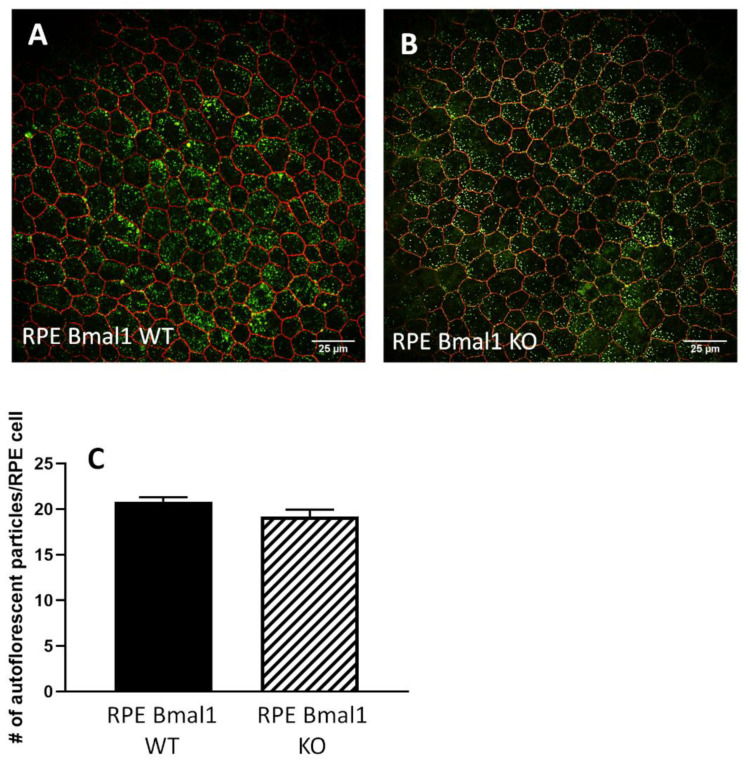
Autofluorescent particle accumulation is not affected by disruption of the RPE circadian clock. We sought to measure the amount of autofluorescence in RPE cells with chronic disruption to the circadian clock in RPE cells. RPE sheets prepared as a flat mount, we quantified the number of apical autofluorescent particles in isolated RPE cells in both old RPE Bmal1 WT mice and RPE Bmal1 KO mice (**A**,**B**). No differences were observed in the number of accumulated apical autofluorescent particles as a result of the loss of the RPE circadian clock (**C**; *p* > 0.1, unpaired *t*-test, *n* = 5–6/group). Data are expressed as mean ± SEM.

## Data Availability

The RNA-seq raw and processed data presented in this publication have been deposited in NCBI’s Gene Expression Omnibus and are openly accessible through GEO Series accession number GSE191104 (Available online: https://www.ncbi.nlm.nih.gov/geo/query/acc.cgi?acc=GSE191104 (accessed on 3 April 2022)). The raw data for SD-OCT and ERG and respective analysis are available upon request to the corresponding author.

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
