# Peer review of "The Circadian Clock in the Retinal Pigment Epithelium Controls the Diurnal Rhythm of Phagocytic Activity"

_ijms, 2022, doi:10.3390/ijms23105302_

Round 1

Reviewer 1 Report

In this manuscript the Authors study the effect of tissue-specific removal of the circadian clock gene Bmal1 in retinal pigment epithelium (RPE) on a variety of morphological and physiological parameters of the retina including the phagocytic activity of RPE, dopamine receptor 2 (D2R) protein expression, daily transcriptome of RPE, retinal thickness, photopic and scotopic electroretinogram (ERG), visual acuity and RPE morphology, some of these in both young and old animals.   

The purpose is legit, the research is important, the methodology is appropriate. The results are presented in a nicely organized manner. I have two major comments:

  1. The breeding strategy for Bmal1 KO mice is not detailed. The figure illustrating it is particularly worrisome: crossing an RRE -/-; Bmal1 fl/fl  and a RpE Cre/Cre; Bmal1 +/+ will result RPE Cre/-;Bmal1 fl/+ mice in F1 (i.e. mice heterozygote for Bmal1 in RPE) , not the RPE Cre/-;Bmal1 fl/fl  (i.e. RPE Bmal1 KO) as indicated in Fig 1A. Sufficient explanation of the breeding strategy is not included in ref 9 either, beyond stating that the transgenics were backcrossed to/ bred on C57BL/6 background. Along these lines, the patchy Bmal1 immuno obtained in the RPE Bmal1 KO (Fig 1E) could indicate a Bmal1 hetrozygote. (Here it is important to note that the figure legend describing Fig 1D and 1E does not give clue what are the red and green signals associated with.) More importantly: if the breeding strategy was indeed a single step crossing as indicated by the figure, working on cell/tissue specific Bmal1 heterozygotes might explain the mostly negative results presented here.
  1. Figure 2 show the data related to the title of the manuscript (“The RPE circadian clock controls the diurnal peak in phagocytosis of POS.”). The data is analyzed with one-way ANOVA, followed by Tukey post hoc (within a genotype?), and revealed significant differences across the different time points. I am wondering why two-way ANOVA (genotype X time) was not used here? Without appropriate statistical analysis I cannot see how the statement (for the title of Figure 2 as well as for the manuscript) came about, even if the figure legend (as well as the corresponding section of the Results. Ln. 106) states that “the amplitude of the diurnal peak (i.e., ZT1) in the number of phagosomes in RPE Bmal1 KO mice was dramatically dampened”, especially Fig 1F shows no difference in the number of phagosomes across the 4 genotypes. Clarification is needed here.

Author Response

Reviewer #1

  1. The breeding strategy for Bmal1 KO mice is not detailed. The figure illustrating it is particularly worrisome: crossing an RRE -/-; Bmal1 fl/fl  and a RpE Cre/Cre; Bmal1 +/+ will result RPE Cre/-;Bmal1 fl/+ mice in F1 (i.e. mice heterozygote for Bmal1 in RPE) , not the RPE Cre/-;Bmal1 fl/fl  (i.e. RPE Bmal1 KO) as indicated in Fig 1A. Sufficient explanation of the breeding strategy is not included in ref 9 either, beyond stating that the transgenics were backcrossed to/ bred on C57BL/6 background. Along these lines, the patchy Bmal1 immuno obtained in the RPE Bmal1 KO (Fig 1E) could indicate a Bmal1 hetrozygote. (Here it is important to note that the figure legend describing Fig 1D and 1E does not give clue what are the red and green signals associated with.) More importantly: if the breeding strategy was indeed a single step crossing as indicated by the figure, working on cell/tissue specific Bmal1 heterozygotes might explain the mostly negative results presented here.

Further detail of the breeding strategy and genoptyping is now included in the manuscript (lines 83-97):  “Rpe-CRE  (Vmd2-rtTA/TetO-cre )23 and Bmal1fl/fl (Jackson laboratory Cat# 007668) were bred to generate the conditional and inducible Cre recombinase expression in retinal pigment epithelium. Mice hemizygous for the Cre locus and homozygous for Bmal1fl/fl locus were crossed to Bmal1fl/flmice to generate mice that were homozygous for the Bmal1 locus and either hemizygous for the CRE locus (Rpe-CRE; Bmal1fl/fl) or non-carrier Bmalfl/fl. The offspring from these mice were used for experimental purposes once genotyped via qPCR for CRE and PCR for Bmal1.  The following qPCR primer pairs and probes were used to genotype the Cre locus (CRE: Forward: 5’-CATCGCTCGACCAGTTTAGTT-3’, Reverse: 5’- CTGACGGTGGGAGAATGTTAAT-3’, Probe: 5'-CGCAGGTGTAGAGAAGGCACTTAGC(FAM)-3', off target control: Forward: 5'-AGGCTTGTCACCCTTCTTTC-3', Reverse: 5'-TAATACTGGTGCTCACAAGGC-3', Probe: 5'-ATGCTGACATCTTGGGCACAGACA(HEX)-3') and the following PCR primers for the Bmal1fl/fl locus (Forward: 5’-TCCTGGTTGGTCCAAGAATATG-3’, Reverse: 5’- CTGACCAACTTGCTAACAATTA-3’; Knockout forward: 5'ACTGGAAGTAACTTTATCAAACTG-3'). Mice that showed systemic knockout of the Bmal1 locus due to leaky expression of the Rpe-CRE construct were not used for experiments or further breeding.  Rpe-CRE; Bmal1fl/fl mice were fed either regular mouse lab diet (RPE Bmal1 WT; LabDiet 5001 Cat# 0001319) or doxycycline (dox; RPE Bmal1 KO; dox; Bioserv Cat# S3888) mouse lab diet at p60-p74.”

We have fixed the figure legend for figure 1 to better describe what the colors on the microphotographs show. The figure legend now reads “Representative microphotographs displaying Bmal1 immunoreactivity (green) in RPE Bmal1 KO and RPE Bmal1 WT mice as prepared in an RPE flat mount with RPE cells demarcated with zonula occludens (red; C-D).” lines 352-354.

  1. Figure 2 show the data related to the title of the manuscript (“The RPE circadian clock controls the diurnal peak in phagocytosis of POS.”). The data is analyzed with one-way ANOVA, followed by Tukey post hoc (within a genotype?), and revealed significant differences across the different time points. I am wondering why two-way ANOVA (genotype X time) was not used here? Without appropriate statistical analysis I cannot see how the statement (for the title of Figure 2 as well as for the manuscript) came about, even if the figure legend (as well as the corresponding section of the Results. Ln. 106) states that “the amplitude of the diurnal peak (i.e., ZT1) in the number of phagosomes in RPE Bmal1 KO mice was dramatically dampened”, especially Fig 1F shows no difference in the number of phagosomes across the 4 genotypes. Clarification is needed here.

We changed the wording of the results section for figure 2 to now read “Figure 2A shows a representative immunocytochemical image of rhodopsin-positive phagosomes in the RPE.   Conditional deletion of Bmal1 from the RPE blunted the daily diurnal phagocytic peak at ZT1 when compared to ZT23 and ZT3 (Figure 2C, p > 0.1, two-way ANOVA, Tukey post-hoc). The nocturnal phagocytic peak at ZT14 was not affected by Bmal1 removal from the RPE when compared to both ZT11 and ZT17 (Figure 2C, # = p < 0.05, two-way ANOVA, Tukey post-hoc).  In contrast, removal of Bmal1 from the retina did not affect the diurnal peak of RPE phagocytic activity at ZT1 when compared to ZT23 and ZT3 (Figure 2E, * = p < 0.05, two-way ANOVA, Tukey post-hoc) or the nocturnal peak at ZT14 when compared to ZT11 (Figure 2E, # = p < 0.05, two-way ANOVA, Tukey post-hoc).  Wild-type controls for both the retina and RPE maintained both the morning phagocytic peak (Figures 2B and 2D, * = p < 0.05, two-way ANOVA, Tukey post-hoc) and nocturnal phagocytic peak (Figures 2B and 2D, # = p < 0.05, two-way ANOVA, Tukey post-hoc). Interestingly, there were no differences in the total number of phagosomes engulfed in all mouse models used (Figure 2F; p > 0.1, one-way ANOVA). We were also interested in examining genotype differences at each timepoint sampled. At most times of the day, the greatest differences amongst the three groups where observed for the RPE Bmal1 KO (Figure 2G, * = p < 0.05, two-way ANOVA, Tukey post-hoc). Lines 255-269.

Figure legend for figure 2 was also changed. And now reads “We then performed various phagocytic peak analyses to determine whether ablation of Bmal1 in the retina or RPE has any effect on the phagocytic capacity of RPE cells. The diurnal phagocytic peak at ZT1 was dampened in RPE Bmal1 KO mice (C) while maintained in RPE Bmal1 WT, Retina Bmal1 WT and Retinal Bmal1 KO mice when compared to ZT23 and ZT3 (B, D, E; * = p < 0.05, two-way ANOVA, Tukey post-hoc). Interestingly, the nocturnal peak observed at ZT14 was maintained in RPE Bmal1 WT, RPE Bmal1 KO, and Retinal Bmal1 WT when compared to ZT11 and ZT17 (B-C; # = p < 0.05, two-way ANOVA, Tukey post-hoc). However, this nocturnal peak was only partially observed in Retinal Bmal1 KO mice when compared to ZT11 (E; # = p < 0.05, two-way ANOVA, Tukey post-hoc) and not ZT17 (E; p > 0.1, two-way ANOVA, Tukey post-hoc). No difference was observed in the total number of phagosomes engulfed through the 24 hours among the different genotypes (F; p > 0.1, one-way ANOVA). We then did a group analysis that assessed the phagocytic capacity of Retinal Bmal1 WT, Retinal Bmal1 KO, and RPE Bmal1 WT when compared to RPE Bmal1 KO. Interestingly, all three genotypes demonstrated different phagocytic capacity when compared to RPE Bmal1 KO mice in particular during the daytime period (G; * = p < 0.05, two-way ANOVA, Tukey post-hoc).” lines 381-393.

Reviewer 2 Report

In this manuscript, DeVera et al. studied the role of circadian clock in the retina or RPE in controlling the diurnal phagocytic peak using an RPE tissue-specific Bmal1 KO mice model. The authors found that the daily peak of phagocytic activity is controlled by the circadian clock in the RPE, not in the retina. However, disruption of functional RPE circadian clock through tissue-specific Bmal1 KO in mice does not show apparent deleterious effects. The topic is important and interesting. I do not have any major concerns, only two minor suggestions:

  1. Could the authors explain more details of the RPE tissue-specific Bmal1 KO mice model used in this research? Possible explanations for the remaining 30% Bmal1 protein? How this 30% remaining Bmal1 will affect the main conclusion?

2. Could the authors discuss more about the differences and limitations of using tissue-specific circadian clock gene disruption vs. germline clock gene disruption?

Author Response

Reviewer #2

  1. Could the authors explain more details of the RPE tissue-specific Bmal1 KO mice model used in this research? Possible explanations for the remaining 30% Bmal1 protein? How this 30% remaining Bmal1 will affect the main conclusion?

Details about our mouse model have been added in the material and methods section of the manuscript (page 5, line 83-97. Although there are several Cre mouse lines that target the RPE, all these lines suffer from various insufficiencies that limit their usefulness (e.g., mosaic patterns of Cre expression in the RPE, inducer-independent – leaky - Cre activity, off-target Cre expression, and Cre-mediated toxicity to the RPE.  Thus, as shown in Figure 1, we believe that such a mosaic pattern of expression observed in the RPE of our KO mice is responsible for the remaining level of Bmal1. We have added a sentence about this issue in the discussion (lines 347-348).  We are offering few possible explanations on how the remaining Bmal1 expression may affect the conclusions (lines: 380-388).

  1. Could the authors discuss more about the differences and limitations of using tissue-specific circadian clock gene disruption vs. germline clock gene disruption?

Please see lines 380-388.

Round 2

Reviewer 1 Report

The changes implemented during the revision address my comments/concerns.